# Dendritic Cell Immunotherapy for Solid Tumors: Advances in Translational Research and Clinical Application

**DOI:** 10.3390/cimb47100806

**Published:** 2025-10-01

**Authors:** Mi Eun Kim, Jun Sik Lee

**Affiliations:** Department of Biological Science, Immunology Research Lab & BK21-Four Educational Research Group for Age-Associated Disorder Control Technology, Chosun University, Gwangju 61452, Republic of Korea; kimme0303@chosun.ac.kr

**Keywords:** dendritic cell immunotherapy, solid tumors, cancer vaccines, tumor microenvironment

## Abstract

Dendritic cells (DCs) are critical antigen-presenting cells that orchestrate the interface between innate and adaptive immunity, making them attractive approaches for cancer immunotherapy. Recent advances in the characterization of DC subsets, antigen delivery strategies, and adjuvant design have enabled the enhancement of DC-based vaccines for solid tumors. Clinical studies across melanoma, glioblastoma, prostate cancer, and non-small cell lung cancer have demonstrated safety and immunogenicity, with encouraging signals of clinical efficacy, particularly when DC vaccination is combined with immune checkpoint blockade or personalized neoantigen approaches. However, translational barriers remain, including the immunosuppressive tumor microenvironment, inefficient DC migration, and variability in manufacturing protocols. Developing solutions such as in vivo DC targeting, biomaterials-based delivery systems, high-resolution single-cell analyses, and artificial intelligence-driven epitope prediction are controlled to overcome these challenges. Together, these innovations highlight the evolving role of DC immunotherapy as a foundation of precision oncology, offering the potential to integrate personalized vaccination strategies into standard treatment paradigms for solid tumors. Therefore, in this review, we specifically focus on these advances in dendritic cell immunotherapy for solid tumors and their translational implications.

## 1. Introduction

Dendritic cells (DCs) are a highly specialized subset of antigen-presenting cells characterized by substantial functional heterogeneity and a pivotal role in coordinating innate and adaptive immunity. Through their ability to capture, process, and present antigens to naïve T cells, DCs act as primary initiators of adaptive immune responses and serve as critical components of immunological surveillance. In relation to cancer, particularly solid tumors, DCs have accumulated considerable interest due to their capacity to elicit tumor-specific cytotoxic T lymphocyte (CTL) responses and modulate the immunosuppressive tumor microenvironment (TME). By cross-presenting tumor antigens and promoting T cell infiltration, DCs contribute to directing the immune landscape of tumors toward an immunologically active state [1,2,3]. The immunological significance of DCs was first recognized in 1973 following their discovery by Ralph Steinman, a breakthrough that redesigned the field of immunology. Subsequent studies have established DCs as the most potent antigen-presenting cells, capable of driving both immune activation and tolerance. Steinman’s foundational work was later honored with the Nobel Prize in Physiology or Medicine in 2011, underscoring the profound clinical implications of DC research [4]. Over the past two decades, substantial progress has been made in elucidating the biology of DC subsets, purifying antigen delivery systems, and engineering DC-based vaccines. These advancements have led to a series of preclinical and clinical studies evaluating the safety, immunogenicity, and therapeutic potential of DC immunotherapy across a broad spectrum of malignancies. The clinical success of Sipuleucel-T, the first FDA-approved vaccine for prostate cancer, highlighted the therapeutic potential of dendritic cell-based strategies and confirmed that T cell-directed tumor vaccines are feasible [5,6,7].

Despite these developments, it is still difficult to bring DC-based treatments into general clinical application. Among the main challenges are the tumor microenvironment’s immunosuppressive qualities, variable vaccine manufacturing procedures, and the insufficient migration of ex vivo-generated DCs to secondary lymphoid organs. The establishment of standardized treatment platforms is further complicated by the diversity in maturation phases and DC subset function [8,9]. Promising solutions to these limitations can be found in recent technical developments. The combination of systems immunology, biomaterials engineering, and single-cell transcriptomics has led to a better understanding of DC biology and function. Moreover, the development of synthetic long peptides, RNA-based antigen delivery, and novel adjuvants such as polyinosinic–polycytidylic acid stabilized with poly-L-lysine and carboxymethylcellulose (poly-ICLC) and stimulator of interferon genes (STING) agonists has enhanced the ability to modulate DC activity with greater precision. Personalized DC vaccines incorporating tumor-specific neoantigens identified through next-generation sequencing are also being explored, with early-phase trials reporting favorable immunological outcomes [8,10,11].

This review aims to provide an overview of recent advances in dendritic cell immunotherapy for solid tumors. We review the development and functions of DC subsets, explore key strategies for vaccine design and antigen delivery, and summarize clinical trial results. We also discuss current limitations and outline future directions to improve the efficacy of DC-based immunotherapy.

## 2. DC Subsets

The DCs comprise a phenotypically and functionally diverse group of antigen-presenting cells that arise from distinct developmental pathways and populate various tissues under both steady-state and inflammatory conditions. In humans, the major DC subsets include conventional dendritic cells (cDCs), plasmacytoid dendritic cells (pDCs), and monocyte-derived dendritic cells (moDCs), each exhibiting specialized roles in antigen presentation and immune modulation. Recent advances in high-dimensional single-cell analyses have further expanded this landscape, revealing additional populations such as dendritic cells type 3 (DC3s) and mature regulatory dendritic cells (mregDCs), thereby highlighting the complexity and plasticity of the DC compartment (Table 1) [12,13,14,15].

Conventional DCs are subdivided into cDC1 and cDC2 subsets based on development, surface marker expression, and transcriptional regulation. cDC1s are defined by expression of CLEC9A, XCR1, and CD141 (BDCA-3), and are specialized in the cross-presentation of exogenous antigens to CD8^+^ T cells via MHC class I pathways, a function central to antitumor immunity [16,17]. These cells depend on transcription factors such as BATF3 and IRF8 for lineage commitment and exhibit a strong capacity to prime CTL and natural killer (NK) cells in the TME. High infiltration of cDC1s into tumor sites has been positively associated with enhanced response to immune checkpoint blockade and improved clinical outcomes in various cancers [18,19,20]. Moreover, cDC1s demonstrate a unique ability to retain tumor antigens within early endosomal compartments, allowing for efficient cross-presentation and sustained CD8^+^ T cell stimulation [21]. In contrast, cDC2s, characterized by markers such as CD1c (BDCA-1) and SIRPα, specialize in presenting antigens to CD4^+^ T cells and promoting a broad range of helper T cell responses. This subset is transcriptionally dependent on IRF4 and Notch signaling, and plays a crucial role in regulating Th1, Th2, and Th17 differentiation. cDC2s are also involved in maintaining immune tolerance and orchestrating tissue-specific immune responses [22,23]. Recent single-cell transcriptomic profiling has identified functional heterogeneity within cDC2s, including pro-inflammatory subsets such as DC3s, which co-express monocyte-associated markers CD14 and CD163 and expand under inflammatory conditions. DC3s are implicated in promoting Th17 polarization and are found at increased frequency in autoimmune diseases and certain tumor settings, where they contribute to a chronic inflammatory environment [24,25,26].

**Table 1 cimb-47-00806-t001:** Functional characteristics of dendritic cell subsets in tumor immunity.

DC Subset	Key Markers	Main Functions	Role in Cancer	Ref.
cDC1	CD141 (human), CD8α, XCR1 (mouse)	Cross-presentation via MHC-I; CD8+ T cell activation	Key in antitumor CTL responses; poor outcomes if absent	[2,16,17]
cDC2	CD1c (human), CD11b (mouse)	MHC-II presentation; CD4+ T cell help; Th2/Th17 responses	Supports adaptive immunity; modulates TME	[22,23]
pDCs	CD123, BDCA-2	Produce type I IFNs; roles in tumor promotion/suppression	Context-dependent; can promote or suppress tumors	[27,28,29]
DC3	CD14, CD163, CD1c	Inflammatory phenotype; hybrid between monocytes and cDCs	Emerging; potential T cell activation under cytokine cues	[25,26]
mregDCs	PD-L1, IL-10, CCR7, LAMP3	Immunoregulatory; linked to immune checkpoint modulation	Potential regulators of checkpoint therapy efficacy	[30,31,32]

Plasmacytoid DCs (pDCs) form a morphologically and functionally distinct lineage that is characterized by high expression of CD123, CD303 (BDCA-2), and CD304 (BDCA-4). These cells are recognized for their ability to produce large amounts of type I interferons (IFN-α/β) in response to viral nucleic acids sensed through Toll-like receptor (TLR) 7 and TLR 9 [27,28,29]. While pDCs are primarily associated with antiviral immunity, their role in cancer is controversial. In some tumor settings, pDCs contribute to immune suppression by promoting regulatory T cell (Treg) expansion and expressing immunoinhibitory molecules like ICOS-L. However, when appropriately activated, they can support antitumor responses by promoting NK cells and CTL activation [33,34].

Monocyte-derived DCs (moDCs) are not typically present under steady-state conditions but can be generated ex vivo from blood monocytes using GM-CSF and IL-4. These cells have been widely utilized in clinical trials of DC vaccines due to their ability of expansion and antigen-loading capacity. MoDCs can be matured with TLR ligands or pro-inflammatory cytokines and pulsed with tumor-associated or neoantigens. Despite their clinical efficacy, moDCs exhibit limited migratory behavior and reduced capacity for cross-presentation when compared to primary cDC subsets, which may limit their effectiveness in inducing antitumor immunity [35,36,37].

MregDCs represent a functionally distinct population characterized by the co-expression of maturation markers (CD40, CD86, and CCR7) and immunoregulatory molecules (PD-L1, IL-10, and CD200) [30]. These cells are more common in tumor-draining lymph nodes and tumor lesions, and they originate from both the cDC1 and cDC2 lineages upon uptake of apoptotic tumor antigens. Although mregDCs can present antigens and have large amounts of MHC class II, they also suppress T cell activation by expressing checkpoint molecules. Their induction is associated with AXL signaling and regulated by cytokines such as IFN-γ and IL-4 [38]. Because mregDCs are both immunogenic and tolerogenic, they are important candidates for therapeutic modification in the tumor microenvironment [31,32].

Our understanding of DC biology in cancer is constantly changing as a result of the growing understanding of DC subset heterogeneity, which is supported by single-cell transcriptomics, proteomics, and computational modeling. Subset-specific targeting will probably be helpful for future therapeutic approaches since it will enable the selective activation of immunostimulatory DCs (like cDC1s) and the reprogramming or depletion of tolerogenic subsets (like mregDCs). It is important for understanding the origin, context-dependent roles, and adaptability of each DC subgroup in order to advance DC-based immunotherapies.

## 3. DC Vaccine Design Strategies

DC-based cancer vaccines aim to stimulate antigen-specific antitumor immune responses by leveraging the potent antigen-presenting capabilities of DCs. The classical ex vivo strategy typically involves isolating autologous moDCs, loading them with tumor-associated antigens (TAAs) or patient-specific neoantigens, inducing their maturation through pro-inflammatory stimuli, and reintroducing them into the patient to initiate T cell-mediated immunity. Among these, moDCs have been the most frequently employed subset due to their accessibility and established culture protocols under clinical-grade (GMP-compliant) conditions [9,39]. Antigen-loading organizations have progressed from the simple pulsing of DCs with synthetic peptides or tumor lysates to more sophisticated approaches such as RNA electroporation, DNA transfection, and fusion with whole tumor cells. RNA-based loading, in particular, allows the expression of multiple epitopes from a single transcript and has demonstrated efficacy in both CD4+ and CD8+ T cell priming, partly due to the activation of intracellular TLR7/8, which enhances DC maturation and type I IFN production [40,41]. Neoantigen vaccines, developed using next-generation sequencing (NGS) and bioinformatic prediction algorithms, offer a high degree of tumor specificity while minimizing the risk of autoimmunity. Personalized neoantigen vaccines have shown promise in early-phase clinical trials by expanding the extensiveness and clonality of tumor-specific T cell repertoires and inducing both dominant and subdominant responses [42,43].

Another promising approach is the in vivo targeting of endogenous DCs using antigen–antibody conjugates. These strategies utilize monoclonal antibodies directed against surface receptors expressed selectively on cross-presenting DCs, such as DEC-205 and CLEC9A, to deliver antigens directly to their intended cellular targets [44,45]. Co-administration of maturation signals, such as poly(I:C) (TLR3 agonist), CD40 agonists, or STING ligands, is essential to ensure appropriate DC activation and prevent tolerance induction [46,47,48]. Additionally, vaccines targeting the XCR1 chemokine receptor, uniquely expressed on cDC1s, have been employed to exploit their superior cross-priming abilities [49,50]. Biomaterials and nanotechnology have enabled the engineering of delivery platforms that enhance antigen presentation and immunogenicity. Lipid nanoparticles, as utilized in mRNA-based COVID-19 vaccines, have demonstrated efficacy in delivering RNA constructs to lymphoid tissues and promoting cytoplasmic translation in antigen-presenting cells [51]. Other platforms include polymeric particles, dendrimers, hydrogels, and implantable scaffolds that can be designed to release antigens and adjuvants in a controlled manner. Biodegradable slow-release depots have been employed to create localized immunostimulatory functions that recruit DCs and enhance T cell priming [51,52,53,54].

The integration of DC vaccines with immune checkpoint inhibitors (ICIs), such as anti-PD-1 and anti-CTLA-4 antibodies, is being actively explored in clinical trials. Preclinical and clinical evidence suggests that the immunostimulatory environment generated by DC vaccination may sensitize tumors to ICIs, resulting in synergistic antitumor effects. Moreover, the combination with cytokines, TLR agonists, or oncolytic viruses is under investigation to further amplify T cell responses and modulate the TME [55,56,57]. Functionally mature DCs producing interleukin (IL)-12p70, a critical cytokine for type 1 T cell polarization, have been associated with enhanced clinical outcomes. Studies have shown that vaccine-induced IL-12p70 production correlates with increased tumor-specific CD8+ T cell responses, improved effector to regulatory T cell ratios, and prolonged progression-free survival. This highlights the importance of optimizing DC maturation protocols, including dual stimulation via CD40 and IFN-γ pathways, or incorporation of TLR ligands to maximize IL-12p70 production [58,59]. Nonetheless, standardization of DC vaccines is still a significant translational challenge. Protocols for DC generation, antigen loading, maturation, and quality control differ considerably between studies, affecting reproducibility and clinical comparability. There are still ongoing efforts to develop phenotypic characteristics and general effectiveness assays that are indicative of in vivo efficacy (Table 2).

Therefore, advances in neoantigen identification, targeted delivery platforms, adjuvant systems, and combinatorial strategies are collectively enhancing the therapeutic potential of DC-based vaccines. DC immunotherapy has the potential to be a key component of the individualized treatment of solid tumors as clinical trials keep developing these approaches.

## 4. DC-Based Clinical Trials

Clinical trials have played a central role in advancing the understanding and therapeutic application of DC-based vaccines in cancer immunotherapy [69]. In the last twenty years, multiple phase I, II, and limited phase III trials have examined the safety, immunogenicity, and clinical efficacy of DC vaccines targeting various malignancies such as melanoma, glioblastoma, prostate cancer, and non-small cell lung cancer [70,71,72] (Table 3).

Melanoma has been one of the most extensively studied cancers in this background. In a seminal phase I study by Carreno et al., personalized neoantigen-pulsed DC vaccines led to a significant expansion in the repertoire and clonal diversity of tumor-specific CD8+ T cells in advanced melanoma patients [88]. The vaccine induced potent and sustained T cell responses to both dominant and subdominant neoantigens, which were confirmed via MHC-multimer staining and functional assays. Importantly, T cell repertoire diversity increased after vaccination, revealing the potential of DC-based strategies to reshape antitumor immunity at a molecular level [89]. In another melanoma study, Wilgenhof et al. employed TriMixDC-MEL, a vaccine platform comprising mRNA-electroporated DC encoding CD40L, CD70, and constitutively active TLR4, combined with ipilimumab (anti-CTLA-4). This combination yielded objective response rates in 38% of metastatic melanoma patients, including complete responses. These findings support the notion that rational combinations of DC vaccination with ICIs can overcome T cell exhaustion and enhance therapeutic efficacy [89].

In glioblastoma, the DCVax-L phase III trial conducted by Liau et al. demonstrated a survival benefit using autologous tumor lysate-loaded DCs. Patients receiving DCVax-L achieved a median overall survival of nearly two years compared to retrospective controls. In a separate study by Xie et al., RNA-loaded DCs delivering personalized neoantigens induced intratumoral CD8+ T cell infiltration, with evidence of clonal replacement and prolonged survival in a subset of patients. These trials collectively indicate that even in immunologically “cold” tumors like glioblastoma, DC vaccines can modulate the tumor immune microenvironment and provide clinical benefit [60,73].

The development of DC immunotherapy has been mostly based on prostate cancer, as demonstrated by sipuleucel-T, the first autologous DC-based cellular therapy to receive FDA approval. In the IMPACT trial, sipuleucel-T extended median overall survival by approximately four months in men with metastatic castration-resistant prostate cancer, without significantly impacting tumor burden. The vaccine consists of autologous peripheral blood mononuclear cells, including APCs activated ex vivo with a fusion protein of prostatic acid phosphatase (PAP) linked to GM-CSF [75,76,77].

Non-small cell lung cancer has also been a focus of clinical innovation. A Japanese phase I trial using moDCs pulsed with MUC1 peptides demonstrated safety and immunologic activity, although tumor responses were limited [90,91]. More recent studies integrating DC vaccination with PD-1 blockade have shown enhanced antigen-specific T cell proliferation, with early clinical signals of partial tumor regression. These results suggest a potential role for DC vaccines as sensitizers for checkpoint blockade [71,83,84].

Despite their promise, DC-based vaccines face several limitations. Objective response rates remain limited, and variability in clinical outcomes is influenced by factors such as patient immune status, tumor antigen heterogeneity, and TME immunosuppression. Additionally, inefficient DC migration to lymphoid organs and inconsistent manufacturing protocols complicate general clinical application. To address these challenges, ongoing trials are incorporating synergistic modalities such as TLR agonists, oncolytic viruses, and cytokines like GM-CSF and FLT3L to enhance in vivo DC activation and recruitment. Furthermore, “off-the-shelf” allogeneic DC products, including those derived from induced pluripotent stem cells (iPSCs), are under development to improve scalability and reduce practical barriers associated with autologous platforms [92,93,94,95]. The future of DC vaccine trials will likely highlight biomarker integration to predict responders, refine treating schedules, and monitor longitudinal immune responses. The management of vaccine manufacturing and quality control protocols across trial sites is essential to facilitate large-scale phase III validation studies and regulatory approval.

## 5. Challenges and Future Perspectives

Despite remarkable advancements in DC-based immunotherapies, their clinical translation to widespread use in solid tumors remains constrained by complicated biological and practical challenges. Among the most pressing biological limitations is the immunosuppressive nature of the TME, which actively impairs DC maturation, antigen presentation, and subsequent T cell priming. Tumor-secreted cytokines and growth factors such as IL-10, TGF-β, VEGF, and prostaglandin E2 interfere with DC function, promote Treg expansion, and attract myeloid-derived suppressor cells (MDSCs), all of which blunt antitumor immunity [96,97,98,99]. Another critical hurdle is the inefficient migration of ex vivo-generated DCs to secondary lymphoid organs. Following administration, only a minority of injected DCs reach the draining lymph nodes, severely restricting their ability to activate naïve T cells. Strategies to address this include chemokine engineering, such as upregulating CCR7 expression on DCs or co-administering CCL19/CCL21 to enhance lymph node homing [100,101]. The lack of standardization in DC vaccine production further complicates clinical arrangement. There is significant variability in antigen-loading protocols (e.g., peptide pulsing, tumor lysate, RNA transfection), maturation stimuli (e.g., TLR ligands, CD40L, cytokines), and release conditions (e.g., IL-12p70 secretion, costimulatory molecule expression), all of which impact product quality and immunogenicity. Current efforts focus on the development of automated, GMP-compliant platforms, and standardized antigen libraries could facilitate more consistent production and broader accessibility [102,103]. Patient-specific factors, such as immune senescence in elderly populations, tumor burden, and prior chemotherapy or corticosteroid exposure, can negatively affect DC function and T cell responsiveness. These inter-individual variables have led to the increasing importance of biomarker-driven personalization. More efficient treatment might become possible by using predictive markers such as tumor mutational burden (TMB), HLA haplotype diversity, immune gene signatures, and the composition of tumor-infiltrating lymphocytes (TILs) [104,105]. In vivo targeting of DCs offers a promising solution to circumvent the need for labor-intensive ex vivo manipulation. Strategies under investigation include the use of receptor-specific ligands, bispecific antibodies, and nanoparticle-based delivery systems that target DEC-205, CLEC9A, or other surface receptors preferentially expressed on cross-presenting DCs. These approaches also allow for simultaneous delivery of antigens and adjuvants to promote immunogenic maturation and efficient T cell priming [106,107]. Single-cell RNA sequencing and geographic transcriptomics are examples of high-resolution technologies that have led to a better knowledge of DC subset dynamics in the TME. These approaches have revealed specialized regulatory DCs characterized by high expression of immunosuppressive mediators such as PD-L1, IL-10, and IDO. Efforts to reprogram these DCs toward an immunostimulatory phenotype using small molecules, metabolic modulators, or genetic engineering are proceeding [30,108,109,110].

Artificial intelligence (AI) and machine learning are increasingly being integrated into DC vaccine development. Algorithms are being used to predict high-affinity neoepitopes, optimize antigen processing predictions, and model immune interactions at the systems level. Platforms such as MHCflurry enable rapid and accurate prediction of peptide–MHC binding affinity across diverse alleles, streamlining the personalization of DC vaccines [109,111,112]. Ultimately, achieving strong clinical responses will require strategies that support long-term immune memory. DC-derived cytokines such as IL-12, IL-15, and type I interferons play critical roles in the differentiation and maintenance of stem-like memory and tissue-resident memory T cells. Sustained delivery of these cytokines through engineered DCs or synthetic scaffolds may promote prolonged antitumor surveillance. These challenges and the corresponding strategies currently being explored are summarized in Figure 1, which provides a visual overview of the key obstacles to DC-based vaccine efficacy and the innovative solutions under investigation.

Recent advances in integrated proteogenomic and immunopeptidomic analyses enable direct identification of tumor peptides that are actually presented on HLA molecules, improving the precision of neoantigen selection for DC vaccines. Mass spectrometry-based immunopeptidomics coupled with tumor exome or transcriptome sequencing has recovered mutant and noncanonical peptides that cannot be predicted from DNA sequence alone, and pipelines that integrate these data have been shown to prioritize candidate neoantigens for personalized vaccines [113,114]. Proteogenomic workflows that combine RNA sequencing, whole exome sequencing, and deep proteomics increase confidence in antigen discovery by confirming protein level expression of variant peptides and by detecting peptides derived from noncoding or noncanonical sources; such confirmation reduces false positive neoantigen calls and guides selection of peptides loaded onto DCs [115]. Moreover, training the innate immune compartment can improve antigen presentation and adaptive responses; experimental and clinical studies document that certain stimuli induce long-lasting functional and epigenetic reprogramming in DCs, which enhances cytokine production and antigen-presenting capacity on subsequent challenge. Strategies that intentionally induce trained innate states with defined ligands or adjuvants have been shown to augment DC-mediated T cell priming in preclinical cancer models and are being evaluated for clinical translation [116]. Spatially resolved transcriptomic and proteomic profiling permits high-resolution mapping of where different immune subsets, including DCs, localize relative to tumor cells and stromal compartments; these methods have revealed spatial patterns that correlate with immune activation or suppression and can identify microanatomical niches where vaccine-delivered DCs would most likely encounter T cells or be inhibited by suppressive cells. Integrating spatial data into vaccine design can inform route of administration, adjuvant choice, and combinations intended to relieve local suppression [117]. Therefore, proteogenomic confirmation of candidate neoantigens, induction of trained innate programs that enhance DC function, and spatial multiomics that define permissive versus suppressive tissue microenvironments are evidence-based strategies that can be directly applied to refine antigen selection, improve DC vaccine immunogenicity, and rationally design combination regimens for solid tumors [116].

## 6. Conclusions

DC-based immunotherapy has developed as a promising and advanced solid tumor therapeutic approach, capitalizing on the unique ability of DCs to bridge innate and adaptive immunity through efficient antigen presentation and T cell priming. Over the past decade, advances in our understanding of DC subsets, vaccine design technologies, and immunomodulatory adjuvants have markedly improved the precision and immunogenicity of DC-based cancer vaccines. Clinical studies have consistently demonstrated the safety and immunogenic potential of DC vaccines, with select trials reporting objective responses and survival benefits particularly when combined with immune checkpoint inhibitors or personalized neoantigen platforms. Despite these advances, several obstacles remain. The immunosuppressive TME, limited DC migration to lymphoid tissues, and variability in patient immune competence continue to restrict the efficacy and scalability of DC immunotherapy. To overcome these challenges, new approaches like in vivo DC targeting, single-cell transcriptomics, RNA profiling, and AI-based epitope prediction are being applied to vaccine development. These tools are qualifying more refined antigen selection, subset-specific DC modulation, and personalization of immunotherapeutic treatments. Additionally, biomarker-driven patient stratification and standardized manufacturing protocols will be essential for regulatory approval and clinical application. In the future, work must concentrate on improving DC migratory and functional characteristics, creating combination therapies that work in concert with the tumor immune environments, and evoking strong memory T cell responses. Combination with currently used therapeutic methods, like cytokine therapies, oncolytic viruses, and checkpoint inhibition, may increase efficacy even further.

Therefore, DC-based cancer immunotherapy is at a critical turning point in its development. DC vaccines have the potential to advance from experimental approaches into key elements of precision cancer therapy for solid tumors with ongoing interdisciplinary research, clinical advancement, and translational investment.

## Figures and Tables

**Figure 1 cimb-47-00806-f001:**
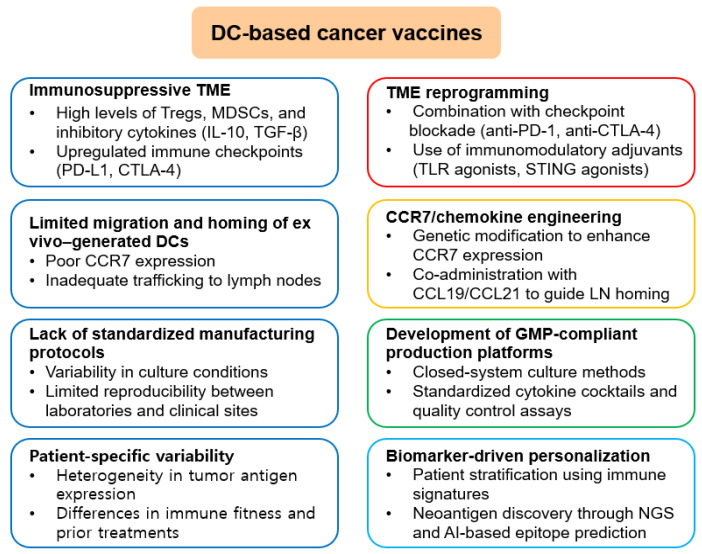
Expanded challenges and potential solutions for DC-based cancer vaccines. A schematic illustration of the major obstacles hampering the efficacy of DC-based immunotherapy and corresponding strategies to overcome them. The immunosuppressive tumor microenvironment, characterized by Tregs, MDSCs, inhibitory cytokines, and checkpoint molecules, can be reprogrammed using checkpoint blockade or immunostimulatory adjuvants. Limited DC migration and homing due to poor CCR7 expression may be addressed by genetic engineering and chemokine co-administration. Lack of standardized manufacturing protocols requires GMP-compliant closed culture systems and validated cytokine/quality control methods. Finally, patient-specific variability arising from antigen heterogeneity and immune fitness differences can be mitigated by biomarker-driven personalization and neoantigen-based vaccine design.

**Table 2 cimb-47-00806-t002:** Approaches to dendritic cell-based cancer vaccines.

Strategy	Description	Current Status	Ref.
Ex vivo-loaded moDCs	Patient-derived monocytes differentiated into DCs and loaded with tumor antigens ex vivo.	Most widely used in early trials; proven safety	[1,9,39]
Neoantigen-based vaccines	Tumor-specific mutations identified and used to generate personalized vaccine epitopes.	Highly personalized, ongoing phase I/II trials	[42,43,60,61]
RNA/mRNA-based vaccines	Synthetic mRNA encoding tumor antigens delivered into DCs or injected for uptake by endogenous DCs.	Promising efficacy; active development post-COVID-19	[51,62,63,64]
In vivo targeting (DEC-205, CLEC9A)	Targeting antigens directly to DC subsets using antibodies against DC-specific receptors.	Increased targeting specificity; in preclinical/early trials	[44,45,65]
Biomaterial/nanoparticle delivery	Encapsulation of antigens/adjuvants in nanoparticles for co-delivery and controlled release.	Improves delivery efficiency; active preclinical stage	[51,52,53,54,66,67]
DC with checkpoint inhibitors	DC vaccination combined with PD-1/PD-L1 or CTLA-4 blockade to enhance T cell activation.	Synergistic combinations under active evaluation	[55,56,57,68]

**Table 3 cimb-47-00806-t003:** Clinical trials of dendritic cell-based vaccines in solid tumors.

Cancer Type	Vaccine Type	Clinical Phase	Outcomes	Ref.
Glioblastoma	DCs loaded with tumor lysate	Phase III	Improved PFS; OS limited	[60,73,74]
Prostate cancer	Sipuleucel-T (FDA-approved)	Phase III	Improved OS in select patients	[75,76,77,78]
Melanoma	Neoantigen-pulsed DCs	Phase I/II	T cell activation; early signs of efficacy	[79,80,81,82]
Non-small cell lung cancer	RNA-modified DCs	Phase I	Well-tolerated, promising immune response	[71,83,84,85]
Head and neck squamous cell carcinoma	DCs pulsed with HPV E6/E7	Phase II	Good safety profile, immune activation	[86,87]

## Data Availability

No new data were created or analyzed in this study. Data sharing is not applicable to this article.

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
