# Peer review of "Dendritic Cell Immunotherapy for Solid Tumors: Advances in Translational Research and Clinical Application"

_cimb, 2025, doi:10.3390/cimb47100806_

Round 1
Reviewer 1 Report
Comments and Suggestions for Authors
In the manuscript entitled “Dendritic Cell Immunotherapy for Solid Tumors: Advances in Translational Research and Clinical Application”, the authors discussed the translational relevance of dendritic cell-based immunotherapy strategies in solid tumors. Below are the comments to improve the manuscript.
- The authors should use more flow charts, graphical representations and diagrams to depict different issues or challenges facing DC-based anti-cancer vaccines and their corresponding potential solutions/strategies.
- The authors should also discuss relevant proteogenomic, trained immunity, spatial transcriptomics/proteomics strategies in the context of DC-based anti-cancer vaccines against solid tumors.
Author Response
Response for Reviewer I
In the manuscript entitled “Dendritic Cell Immunotherapy for Solid Tumors: Advances in Translational Research and Clinical Application”, the authors discussed the translational relevance of dendritic cell-based immunotherapy strategies in solid tumors. Below are the comments to improve the manuscript.
Comment 1: 1. The authors should use more flow charts, graphical representations and diagrams to depict different issues or challenges facing DC-based anti-cancer vaccines and their corresponding potential solutions/strategies..
Response: Thank you for your helpful comment. As suggested by the reviewer, we have incorporated a newly created Figure 1 into the manuscript, which provides a schematic overview of the key challenges associated with DC-based anti-cancer vaccines and highlights the corresponding potential solutions and strategies
Comment 2: The authors should also discuss relevant proteogenomic, trained immunity, spatial transcriptomics/proteomics strategies in the context of DC-based anti-cancer vaccines against solid tumors.
Response: We sincerely appreciate the reviewer’s valuable suggestion. We have added a discussion in the manuscript on relevant proteogenomic approaches, trained immunity, and spatial transcriptomic/proteomic strategies in the context of DC-based anti-cancer vaccines for solid tumors.

Reviewer 2 Report
Comments and Suggestions for Authors This is an interesting review manuscript, and it presents an acceptable organization regarding the topics it addresses. These are the observations regarding the formatting:• On line 63, it is advisable to include the meaning of the acronyms.
• On line 87, review the wording because the "on" is repeated.
• On lines 93-94, review the formatting.
• For Table 1, considering that the text first explains DC3, it would be good to place it before pDC.
In the "Challenges and Future Perspectives" section, the use of computational or artificial intelligence strategies for epitope prediction, such as MHCflurry, could be improved, as these types of tools focus on the development of immunoproteomics.
Author Response
Response for Reviewer II
This is an interesting review manuscript, and it presents an acceptable organization regarding the topics it addresses. These are the observations regarding the formatting
Comment 1:
- On line 63, it is advisable to include the meaning of the acronyms.
- On line 87, review the wording because the "on" is repeated.
- On lines 93-94, review the formatting.
- For Table 1, considering that the text first explains DC3, it would be good to place it before pDC.
In the "Challenges and Future Perspectives" section, the use of computational or artificial intelligence strategies for epitope prediction, such as MHCflurry, could be improved, as these types of tools focus on the development of immunoproteomics.
Response: Thank you for your helpful comment. We have revised the manuscript in accordance with the reviewer’s request.
